# *HLA-DRB1*14:54* Is Associated with Pulmonary Alveolar Proteinosis: A Retrospective Real-World Audit

**DOI:** 10.3390/biomedicines11112909

**Published:** 2023-10-27

**Authors:** Mengqian Li, Qinglin Liu, Weiwen Wang, Lili Jiang

**Affiliations:** Department of Pathology, West China Hospital, Sichuan University, Chengdu 610041, China; limengqian@scu.edu.cn (M.L.); lqll2023@126.com (Q.L.); wangww@wchscu.cn (W.W.)

**Keywords:** pulmonary alveolar proteinosis, autoimmune diseases, *HLA-DRB1*14:54*, *HLA-DRB1*08:03*

## Abstract

Background: Pulmonary alveolar proteinosis (PAP) is a rare pulmonary disease characterized by abnormal accumulation of pulmonary surfactant lipids in alveoli or terminal bronchioles, leading to increased infection risk and progressive respiratory failure. Approximately more than 90% of all cases are autoimmune PAP (aPAP). Since one of the predisposing factors has been identified as genes located within the major-histocompatibility-complex region, an investigation of *human leukocyte antigen* (*HLA*) alleles associated with the risk of aPAP is warranted. Methods: We retrospectively studied 60 patients pathologically diagnosed with PAP from 2019 to 2022. Patients were divided into the aPAP group or secondary PAP (sPAP) group according to their clinical information. Qualified DNA was extracted from the paraffin-embedded tissue of 28 patients, and the PCR-sequence-based typing method was used for *HLA-DRB1* genotyping. Results: A similar *HLA-DRB1* allele profile (including the *HLA-DRB1*08:03*) between the aPAP group and sPAP group was revealed, except that *HLA-DRB1*14:54*, which has never been reported in aPAP patients, was only detected in the aPAP group rather than the sPAP group (19.4% vs. 0.0%, *p* = 0.030). Under inhaled granulocyte-macrophage colony-stimulating factor therapy, more clinical remission was observed in *HLA-DRB1*14:54* carriers rather than in *HLA-DRB1*08:03* carriers (80.0% vs. 57.1%). Conclusions: Our real-world study revealed for the first time that a population with *HLA-DRB1*14:54* was subject to aPAP, and *HLA-DRB1*14:54* might imply a response in aPAP patients to inhaled granulocyte-macrophage colony-stimulating factor in aPAP patients.

## 1. Introduction

Pulmonary alveolar proteinosis (PAP), which was first described in 1958, is a rare diffuse lung disease with a prevalence of 6–7 cases per million population worldwide. This disease is characterized by the abnormal accumulation of phosphor-lipoproteinaceous material pulmonary surfactant lipids within alveoli, which could cause restrictive pulmonary dysfunction, decreased diffusion capacity, and even progressive respiratory failure [1].

Three distinct etiologies have been identified in PAP: primary PAP, secondary PAP (sPAP), and congenital PAP [2]. Primary PAP is caused by the disruption of g granulocyte–macrophage colony-stimulating factor (GM-CSF) signaling and can be further classified as autoimmune PAP (aPAP) and hereditary PAP. The former constitutes more than 90% of all primary PAP, which is caused by high concentrations of GM-CSF autoantibodies that interfere in surfactant catabolism in alveolar macrophages [2,3]. The latter arose from mutations in *CSF2RA* or *CSF2RB* (encoding GM-CSF receptor subunits) [2,3].

Notwithstanding the major advances in our understanding of PAP over the past decades, several important questions remain unanswered. For example, although mutations involving hereditary PAP have been reported as described above [4,5], the genetic basis underlying aPAP has not been thoroughly investigated. The human leukocyte antigen (*HLA*) system plays a crucial role in the regulation of immune system discriminating between “self and non-self” [6]. Given the autoimmune etiology of aPAP, an association between the susceptibility to aPAP and the *HLA* system has long been expected [7]. Recently, a study on the Japanese population provided evidence that *HLA-DRB1*08:03* was significantly associated with an increased risk of aPAP [8]. It is noteworthy that different alleles at the *HLA-DRB1* locus, such as *DRB1*04:04*, *04:01*, *08:03*, *15:01*, and *16:02*, also have shown good association with autoimmune diseases or the production of autoantibodies [6,9,10,11,12]. Whether other *HLA-DRB1* alleles are also involved in aPAP, and whether *HLA-DRB1*08:03* is associated with PAP of other subtypes remain unknown.

Here, we retrospectively collected data from 60 patients diagnosed with PAP in West China Hospital and grouped patients as aPAP and sPAP based on the clinical information. Adopting the PCR-sequence-based typing (PCR-SBT) method, we found *HLA-DRB1*08:03*, did not show a significant difference in frequencies between groups. However, *HLA-DRB1*14:54* was an exception that was only detected in the aPAP group and might be related to higher response rates to inhaled GM-CSF in aPAP patients.

## 2. Methods

### 2.1. Subjects

This is a retrospective single-center study of PAP conducted by the Department of Pathology, West China Hospital of Sichuan University. This study was approved by the ethical committee of West China Hospital of Sichuan University (approval No.20180612). West China Hospital is an affiliated teaching hospital where each patient signs informed consent that supports teaching and scientific use of their specimens before a biopsy.

We analyzed 69 patients with a pathological diagnosis of PAP in West China Hospital from January 2019 to May 2022. Patients were eligible if their lung tissue sections and paraffin-embedded tissues were both in good preservation in the Department of Pathology, and their clinical information and imaging information were collected by the electronic medical record system of West China Hospital. The exclusion criteria included diagnosis by cell smear or special staining of bronchoalveolar lavage fluid, paraffin-embedded tissues in poor preservation, incomplete clinical information, other potential hereditary diseases, and specimens from other hospitals. The diagnosis of PAP was based on transbronchial lung biopsy (N = 41), percutaneous lung biopsy (N = 15), and surgical lung biopsy (N = 4). Pathological reassessment of the lung tissue sections was performed independently by two experienced lung pathologists.

Sixty patients were finally included and divided into the aPAP group (N = 46) or sPAP group (N = 14) as per the presence of primary diseases (such as tumors or infection) after a thorough investigation of the medical history. A total of 29 aPAP patients accepted inhaled GM-CSF therapy with doses ranging from 125 to 300 μg (bid) for from 2 weeks to 3 years, which was at a one-week or one-day interval or with a treatment schedule following the physicians’ recommendations. Patients were regularly evaluated by their physicians at West China Hospital after the GM-CSF inhalation therapy, and adjustments to the GM-CSF doses were made according to patients’ conditions.

### 2.2. Diagnostic Criteria

The diagnosis of PAP was established by the presence of characteristic findings from thoracic high-resolution computed tomography (HRCT) and pathologic specimens obtained by lung biopsy [1,2]. The radiologic features characteristic of PAP on HRCT used here included the presence of a patchy or diffuse pattern of ground-glass opacification (GGO) superimposed on interlobular septal thickening in multiple lobes [13]. The criteria for pathological diagnosis of PAP were diffuse acellular eosinophilic material in the alveoli (with or without cholesterol clefts), which exhibited a positive reaction to periodic acid–Schiff staining independent of amylase treatment [1,14].

Participants were assigned a PAP disease severity score (DSS) based on the presence of symptoms and the degree of reduction in PaO_2_ (both determined at their first visit) determined with the individual breathing room air in the supine position as previously described [1]. Briefly, the categories included DSS 1 = no symptoms and PaO_2_ ≥ 70 mmHg; DSS 2 = symptomatic an PaO_2_ ≥ 70 mmHg; DSS 3 = 60 mmHg ≤ PaO_2_ < 70 mmHg; DSS 4 = 50 mmHg ≤ PaO_2_ < 60 mmHg; DSS 5 = PaO_2_ < 50 mmHg. Qualifying symptoms include dyspnea or cough related to PAP.

### 2.3. DNA Extraction and PCR Amplification

For each case, DNA was extracted from formalin-fixed paraffin-embedded tissue using QIAamp DNA FFPE Tissue Kit (Cat. No.: 56404, Qiagen, Hilden, Germany) as previously described [15]. DNA concentrations and purity (A260/A280 ratio) were measured by NanoDrop OneC UV-Vis spectrophotometer (Thermo Fisher Scientific, Waltham, MA, USA). The quality of extracted DNA was further analyzed by electrophoresis in a 1.0% agarose gel and photographed under ultraviolet light.

PCR amplifications were performed as previously described [16]. Primers that amplify the exon 2 of the *HLA-DRB1* gene are shown in Appendix A (provided by the *HLA* Genotyping kit (China Shanghai of Tissuebank, Shanghai, China, http://www.catb.org.cn/three-types-cat/pro1/ (accessed on 21 November 2022)). PCR amplification was confirmed following electrophoresis in a 2.0% agarose gel, and the amplification products were purified using QIAquick Gel Extraction Kit (Cat. No.: 28706X4, Qiagen, Hilden, Germany).

### 2.4. DNA-Sequencing and HLA-DRB1 Genotyping

PCR-SBT method was used for *HLA-DRB1* genotyping. Briefly, the above PCR amplification products were sequenced in a single direction by the ABI 3730XL sequencer (Thermo Fisher Scientific, Waltham, MA, USA). Sequencing primers are shown in Appendix A (provided by the *HLA* Genotyping kit, China Shanghai of Tissuebank, Shanghai, China, http://www.catb.org.cn/three-types-cat/pro1/ (accessed on 21 November 2022)). The sequencing products were compared with the reference sequences in the IPD-IMGT/*HLA* database (https://www.ebi.ac.uk/ipd/imgt/HLA/ (accessed on 21 November 2022)) for *HLA-DRB1* typing.

### 2.5. Statistics

Numeric data were evaluated for a normal distribution using SPSS Statistics software (version 17.0, SPSS Inc, Chicago, IL, USA) and presented as the mean ± SD. Categorical data were presented as a percentage of the total or numerically. *HLA* allele frequencies were calculated by direct counting. For group comparisons, the Student’s *t*-test was used for comparisons between normally distributed data. Comparisons of categorical data were made with Chi-square, Chi-square with Yates’ correction, or Fisher’s exact test. Comparisons of ranked data were made with the Mann–Whitney U test. Analysis was performed using SPSS Statistics software (version 17.0, SPSS Inc, Chicago, IL, USA). All tests were two-sided, and values of *p* < 0.05 were considered statistically significant. Statistical power was analyzed by G*Power software (version 3.1, Faul et al., Germany).

## 3. Results

### 3.1. Study Populations

Sixty patients who fulfilled inclusion criteria were deemed to be eligible subjects and were divided into aPAP group (46 patients,) or sPAP group (14 patients) as per the presence of primary diseases (such as tumors or infection). Thoracic HRCT scans confirmed ground-glass opacification, thickened interlobular septae, thickened intralobular lines, and consolidation in the lungs of both groups (shown in Appendix A).

Characteristics of patients are summarized in Table 1. Two groups showed balanced age distribution and sex ratios at baseline. Clinical manifestations occurred similarly in both groups, except for a significantly higher percentage of chest tightness or chest pain in the sPAP group. We speculate that this might be related to the primary diseases that occurred to sPAP since chest pain often suggests complications, particularly lung infections [17]. Results of the pulmonary function test, arterial blood gas analysis, and disease severity score (DSS) also did not show differences between the two groups. However, the frequency of “Critical or serious illness during hospitalization” was significantly higher in the sPAP group than in the aPAP group. More information is shown in Appendix A.

### 3.2. Histopathological Characteristics

Lung biopsy is considered to be the gold standard for PAP diagnosis [18]. Lung biopsy samples of the 60 patients were reassessed independently by two experienced lung pathologists. Generally, the pulmonary architecture is preserved in most cases. Histopathological characteristics of both groups showed diffuse acellular eosinophilic material harboring cholesterol clefts and particle-laden macrophages in the alveoli (shown in Figure 1A–D). Alveolar septal thickening and minor interstitial inflammation could also be observed in some patients (shown in Figure 1C,D). The eosinophilic material exhibited a positive reaction to periodic acid–Schiff staining independent of diastase digestion (eliminating glycogen), confirming the proteinaceous nature of the intra-alveolar material rather than glycogen (shown in Figure 1E,F). At the same time, the eosinophilic material displayed negative staining by Alcian blue and Gomori methenamine silver, thus also excluding acidic mucins and fungal infections, respectively.

### 3.3. HLA-DRB1 Allelic Polymorphism

To confirm the *HLA-DRB1* allele genotyping of the patients, we adopted the PCR-SBT method with DNAs extracted from paraffin-embedded tissue. Specimens of 18 aPAP patients and 10 sPAP patients offered qualified DNA for PCR amplification and sequencing, which had 36 and 20 *HLA-DRB1* alleles, respectively (Table 2). There was a similar profile of *HLA-DRB1* alleles between both groups from the statistical viewpoint. *HLA-DRB1*08:03* (41.1%, 23/56), *14:54* (12.5.0%, 7/56), and *15:01* (12.5.0%, 7/56) were the most frequent ones in all the patients (Table 2). In aPAP group, *HLA-DRB1*08:03* (25.0%, 9/36) and *14:54* (19.4%, 7/36) constituted the most frequent ones, while *HLA-DRB1*15:01* (25.0%, 5/20), *08:03* (15.0%, 3/20), and *09:01* (15.0%, 3/20) were the most common in sPAP (Table 2). It is worth mentioning that *HLA-DRB1*14:54* has never been reported in PAP, while the frequency of *HLA-DRB1*08:03* in aPAP is in line with the reported data [8].

We further analyzed the proportions of patients carrying *HLA-DRB1*08:03*, *14:54*, and *15:01* alleles in the two groups. We noticed that *HLA-DRB1*14:54* was only detected in the 18 aPAP patients rather than in the 10 sPAP patients (*p* = 0.030), while the other two did not show differences in the proportions between the two groups (Table 3). Thus, *HLA-DRB1*14:54*/*X* genotypes (N = 7, 38.9%), including *HLA-DRB1*08:03*/*14:54*, were also only identified in the aPAP group (N = 18) (shown in Appendix A).

### 3.4. Treatment Response

Inhaled recombinant human GM-CSF therapy is the second-line treatment of aPAP [17]. Clinical trials show that inhaled GM-CSF therapy in aPAP patients is safe and effective, which has been found better than subcutaneous GM-CSF injections [2,17]. In our study, 29 of the 46 aPAP patients accepted inhaled GM-CSF therapy. Clinical remission (per either physicians’ assessment or patient self-assessment) was observed in 100.0% (2/2) of *HLA-DRB1*14:54*/*X* carriers, 50.0% (2/4) of *HLA-DRB1*08:03*/*X* carriers, and 66.7% (2/3) *HLA-DRB1*08:03*/*14:54* carriers, respectively (*X* did not include either *08:03* or *14:54*). In other words, clinical remission (per either physicians’ assessment or patient self-assessment) was observed in 80.0% (4/5) of patients carrying *HLA-DRB1*14:54*, and in 57.1% (4/7) carrying *HLA-DRB1*08:03*, respectively. We speculate that there was an encouraging association between *HLA-DRB1*14:54*/*X* carriage and the clinical response of aPAP patients to inhaled GM-CSF therapy.

## 4. Discussion

We performed a retrospective study to analyze *HLA-DRB1* allelic polymorphism in aPAP and sPAP patients. The representativeness of the patients was satisfactory as the sex ratios, clinical manifestations and signs, imaging manifestations on CT, histopathological characteristics, and the frequency of *HLA-DRB1*08:03* in aPAP group were in line with the previously reported data [8,14,19,20,21]. We found similar *HLA-DRB1* allele frequencies between both groups from the statistical viewpoint. Importantly, *HLA-DRB1*14:54*, which has never been reported in patients with PAP, was only detected in the aPAP group rather than the sPAP group (*p* = 0.030), and *HLA-DRB1*14:54* carriers showed a trend towards higher response rates to inhaled GM-CSF therapy. To our knowledge, this is the first report revealing an underlying genetic predisposition of the carriers of *HLA-DRB1*14:54* towards aPAP.

Because a distinct genetic association to *HLA* alleles is a shared feature among many autoimmune diseases, Anderson et al. compared *HLA*-typed aPAP patients to matched controls to determine if this disease had any *HLA* association. Nevertheless, they did not find any association between the identified 203 alleles (including *HLA-DRB1*08:03*) with aPAP, and *HLA-DRB1*14:54* was not identified in their study [7]. It remains possible that the lack of an association was due to the small sample size (47 patients). Recently, a Japanese study by Sakaue et al. including 198 aPAP patients and 395 control participants first revealed that *HLA-DRB1*08:03* was significantly associated with the genetic risk to the pathogenesis of aPAP, which was also associated with the increased production of anti-GM-CSF antibody [8].

The conclusion made by Sakaue et al. is inconsistent with that by Anderson et al. suggesting a potential clinical utility of *HLA-DRB1* polymorphism in explaining the differences in the severity or treatment response of aPAP. In Sakaue’s study, control participants did not have a past medical history of any immune-related diseases, scilicet a DRB1 allele that was identified as a risk factor unique to aPAP by Sakaue et al. might be shared among different subtypes of PAP, including sPAP. It is noteworthy that *HLA* allele frequency is related to the ancestry populations and geographical regions. *HLA-DRB1*08:03* is an Asian-specific allele [8]. Similar to Sakaue’s findings in the Japanese population [8], our study confirmed that the frequency of *HLA-DRB1*08:03* was 25% in aPAP, which constituted 50% of the tested aPAP patients. However, we did not observe a statistical difference in the proportions of patients with *HLA-DRB1*08:03* between aPAP and sPAP, which might be limited by the small sample size and low statistical power (1-β) and will be hereinafter discussed. This indicated at least that *HLA-DRB1*08:03* carriage was not exclusive to aPAP patients. The allele *HLA-DRB1*14:54*, which was previously classified as *HLA-DRB1*14:01*, is common in the Chinese population [22,23,24]. Studies have shown that *HLA-DRB1*14:54* was significantly positively or negatively associated with some autoimmune diseases [24,25]. Despite its prevalence in the Chinese population, *HLA-DRB1*14:54* was only detected in the aPAP group in our study, supporting a positive relation with this pulmonary autoimmune disease. This is the first report revealing an underlying genetic predisposition of *HLA-DRB1*14:54* carriers towards aPAP.

The aPAP is characterized by large variations in disease severity and clinical prognosis. Whole lung lavage therapy is the standard treatment for PAP, but the symptom was improved for only a short period and required repeated lavages and an invasive procedure, and general anesthesia limits its performance [26]. Inhaled GM-CSF is also an effective therapy for aPAP with easier procedure, longer remission, and fewer side effects than whole lung lavage therapy [20,27], and shows a trend towards better response compared to the subcutaneous route [26]. However, inhaled GM-CSF therapy confers clinical and physiologic benefits in a subset of patients, and the reason for this differential responsiveness is unknown [2]. Identifying patients likely to respond to GM-CSF is a future challenge. In our study, there was a tendency of *HLA-DRB1*14:54* carriers to respond better to inhaled GM-CSF than *HLA-DRB1*08:03* carriers did. We hope carefully controlled prospective studies with a more diverse population and more diverse *HLA* alleles would allow a more conclusive assessment.

This study was subject to some limitations. First, as a result of the retrospective investigation, it was difficult to avoid bias, estimate relative risk, and test serum GM-CSF autoantibody levels of the patients as well. Even so, we could ascertain whether PAP was secondary to other diseases or of congenital or hereditary origins after a thorough check on the medical history of patients. As increasing evidence has shown that there may be an association between inhalational exposure and autoantibody production in the pathogenesis of sPAP [28,29,30,31], environmental exposures should be carefully assessed in PAP patients regardless of the levels of GM-CSF autoantibodies according to Kumar, A.’s opinion [31]. Second, the sample size for our study was limited. The statistical power (1-β) values of analyzing the differences in proportions of patients with *HLA-DRB1*08:03*, *14:54*, and *15:01* between the two groups were 0.168, 0.763, and 0.610, respectively. Because of the non-significant *p* values (>0.05) and low (1-β) values (<0.75), the results from in *HLA-DRB1*08:03* and *15:01* analysis could not reflect true differences between the groups, and non-true negative results might occur. Nevertheless, the value for (1-β) in analyzing *HLA-DRB1*14:54* was 0.763 in the context of *p* < 0.05. We speculate that this value could reach 0.8 in an enlarged sample size. Third, because of the retrospective nature of this study, we could not intervene in the testing and treatment procedures, which led to some nonuniformity that can be found in the adherence to lung function testing, follow-up visits, and GM-CSF therapy. The use of PCR to examine DNA in fixed, paraffin-embedded tissues may be subject to DNA degradation. That resulted in the *HLA-DRB1* data missing in about half of the included patients.

## 5. Conclusions

Our study first reveals that the population with *HLA-DRB1*14:54* was subject to aPAP, and *HLA-DRB1*14:54* might imply a response in aPAP patients to inhaled GM-CSF therapy. Our findings may contribute to the exploration of the underlying mechanism of aPAP. Further study is warranted to determine the association between *HLA-DRB1*14:54* and aPAP.

## Figures and Tables

**Figure 1 biomedicines-11-02909-f001:**
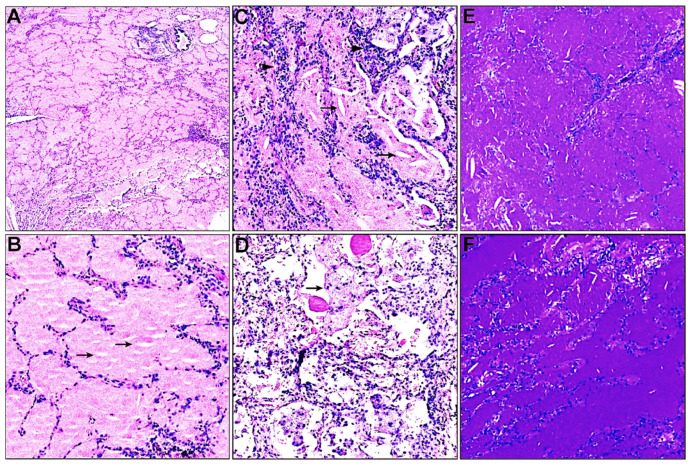
Histopathological characteristics of the lungs of patients with pulmonary alveolar proteinosis. (**A**,**B**) Preserved lung architecture and intra-alveolar accumulation of eosinophilic material harboring cholesterol clefts (arrows) on hematoxylin and eosin stain ((**A**) magnification × 40; (**B**) magnification × 200). (**C**) Minor interstitial inflammation (arrowheads) and intra-alveolar accumulation of cholesterol clefts (arrows) on hematoxylin and eosin stain (magnification × 200). (**D**) Intra-alveolar particle-laden macrophages (arrows) on hematoxylin and eosin stain (magnification × 200). (**E**,**F**) Periodic acid–Schiff stain with ((**E**), magnification × 100) or without diastase digestion ((**F**), magnification × 100).

**Table 1 biomedicines-11-02909-t001:** Clinical characteristics of the patients at baseline.

Variable	aPAP (N = 46)	sPAP (N = 14)	*p* Value
Demographics			
Age at diagnosis, yr ^a^	47.26 ± 10.910	47.14 ± 6.916	0.969
Sex ratio	1.7:1.0	2.5:1.0	
male, no. (%)	29 (63.0)	10 (71.4)	0.7980
female, no. (%)	17 (37.0)	4 (25.6)	
Clinical symptoms, no. (%)			
Cough and/or production of white frothy sputum	29 (63.0)	8 (57.1)	0.690
Dyspnea	21 (45.7)	10 (71.4)	0.091
Hypoxemia	11 (23.9)	7 (50.0)	0.062
Chest tightness or chest pain	2 (4.3)	6 (42.9)	0.001
Respiratory failure	1 (2.2)	2 (14.3)	0.133
FEV1/FVC ^b^, no. (%)			
≥70%	30 (65.2)	6 (42.9)	>0.99
<70%	0 (0.0)	0 (0.0)	
NA	16 (34.8)	8 (57.1)	/
PaO_2_ ^b^, no. (%)			
≥70 mmHg	15 (32.6)	10 (71.4)	0.1578
<70 mmHg	11 (23.9)	2 (14.3)	
NA	20 (43.5)	2 (14.3)	/
SpO_2_ ^b^, no. (%)			
≥94%	17 (37.0)	12 (85.7)	0.502
<94%	2 (4.3)	0 (0.0)	
NA	27 (58.7)	2 (14.3)	/
Disease severity score, no. (%)			
1	4 (8.7)	0 (0.0)	0.393
2	11 (23.9)	10 (71.4)	
3	9 (19.6)	2 (14.3)	
4	2 (4.3)	0 (0.0)	
NA	20 (43.5)	2 (14.3)	/
Critical or serious illness during hospitalization ^c^, no. (%)			
Yes	3 (6.5)	6 (42.9)	0.007
No	36 (78.2)	8 (57.1)	
NA	7 (15.2)	0 (0.0)	/

^a^ Values are expressed as mean ± SD. ^b^ Not all patients accepted the test. ^c^ Some patients were outpatients who only asked for a transbronchial lung biopsy. Statistical analysis was made in the available data. N = the number of patients. aPAP = autoimmune pulmonary alveolar proteinosis; sPAP = secondary pulmonary alveolar proteinosis; FVC = forced vital capacity; FEV1 = forced expiratory volume in 1 s; TLC = total lung capacity; DLCO SB = Diffusing capacity of the lung for carbon monoxide, single-breath method); PaO_2_ = partial pressure of oxygen; SpO_2_ = oxygen saturation; PaCO_2_ = partial pressure of arterial carbon dioxide; pH = potential of hydrogen; NA = not available.

**Table 2 biomedicines-11-02909-t002:** *HLA-DRB1* allele frequencies.

*HLA-DRB1* Alleles	Total (*n* = 56), No. (%)	aPAP (*n* = 36), No. (%)	sPAP (*n* = 20), No. (%)	*p* Value
*08:03*	23 (41.1)	9 (25.0)	3 (15.0)	0.593
*14:54*	7 (12.5)	7 (19.4)	0 (0.0)	0.091
*15:01*	7 (12.5)	2 (5.5)	5 (25.0)	0.091
*16:02*	5 (8.9)	3 (8.3)	2 (10.0)	0.780
*09:01*	5 (8.9)	2 (5.5)	3 (15.0)	0.485
*13:12*	3 (5.3)	3 (8.3)	0 (0.0)	0.479
*10:01*	2 (3.6)	1 (2.7)	1 (5.0)	>0.999
*04:05*	2 (3.6)	1 (2.7)	1 (5.0)	>0.999
*03:01*	2 (3.6)	1 (2.7)	1 (5.0)	>0.999
*11:01*	2 (3.6)	1 (2.7)	1 (5.0)	>0.999
*12:02*	2 (3.6)	1 (2.7)	1 (5.0)	>0.999
*01:01*	1 (1.8)	1 (2.7)	0 (0.0)	>0.999
*04:03*	1 (1.8)	1 (2.7)	0 (0.0)	>0.999
*07:01*	1 (1.8)	1 (2.7)	0 (0.0)	>0.999
*11:06*	1 (1.8)	1 (2.7)	0 (0.0)	>0.999
*12:01*	1 (1.8)	0	1 (5.0)	0.3571
*13:01*	1 (1.8)	0	1 (5.0)	0.3571
*14:05*	1 (1.8)	1 (2.7)	0 (0.0)	>0.999

*n*: the numbers of *HLA-DRB1* alleles. aPAP = autoimmune pulmonary alveolar proteinosis; sPAP = secondary pulmonary alveolar proteinosis.

**Table 3 biomedicines-11-02909-t003:** Proportions of patients with *HLA-DRB1*08:03*, *14:54*, and *15:01* ^#^.

*HLA-DRB1* Alleles	Total (N = 28), No. (%)	aPAP (N = 18), No. (%)	sPAP (N = 10), No. (%)	*p* Value
*08:03*	12 (42.9)	9 (50.0)	3 (30.0)	0.434
*14:54*	7 (25.0)	7 (38.9)	0 (0.0)	0.030
*15:01*	7 (25.0)	2 (11.1)	5 (50.0)	0.067

^#^: each allele was analyzed by Fisher exact tests. N: the number of patients. aPAP = autoimmune pulmonary alveolar proteinosis; sPAP = secondary pulmonary alveolar proteinosis.

## Data Availability

The datasets used and analyzed during the current study available from the corresponding author on reasonable request.

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
