# Peer review of "HLA-DRB1*14:54 Is Associated with Pulmonary Alveolar Proteinosis: A Retrospective Real-World Audit"

_biomedicines, 2023, doi:10.3390/biomedicines11112909_

Round 1

Reviewer 1 Report

HLA-DRB1*14:54 is associated with pulmonary alveolar proteinosis: a retrospective real-world audit is an interesting article; however, I have some observations.

1.       “A total of 29 aPAP patients accepted inhaled GM-CSF therapy with doses ranging from 125 to 300 μg (bid) for from 2 weeks to 3 years”. What factor influences why only 29 patients accept the therapy?

2.       Please indicate how you established the dose and time of therapy of GM-CSF. The literature has several proposals (PMID:32615994 and PMID:20167854)

3.       Table 1. Please include date of FEV1, FVC, TLC, DLCO, PaO2, SpO2

4.       Line 119. “Numeric data were evaluated for a normal distribution using…”; however, authors should use a normality test and after, apply parametric or non-parametric statistics.

5.       Table 2. “aPAP (N = 36) and sPAP (N = 20)”. How is it possible that you now have 20 subjects from the sPAP group if in Table 1 the authors indicate that they have N=14?

6.       Line 195: “Clinical remission (per either physicians' assessment or patient self-assessment)”. Please write the parameters used to evaluate clinical remission.

7.       Line 195: How did you evaluate adherence to treatment?

8.       Line 196: Is the validity of clinical remission results the same if come from the physicians or the patient? Please add criteria and references to support your response.

9.       Line 242-244. Expand this topic in the discussion according to your results

10.   Discussion. The authors mentioned the association of HLA alleles in other populations; however, allele frequency depends on the ancestry population. Can you discuss this point?

Author Response

Dear reviewer:

Thank you for your decision and constructive comments on my manuscript. We have carefully considered the suggestions and tried our best to improve and make some changes to the manuscript. We have made corresponding modifications in red font in the manuscript as you suggested.

Revision notes, point-to-point, are given as follows:

  1. "A total of 29 aPAP patients accepted inhaled GM-CSF therapy with doses ranging from 125 to 300 μg (bid) for from 2 weeks to 3 years". What factor influences why only 29 patients accept the therapy?

Thanks for your comments. We understand your concern about the relatively low acceptability of inhaled GM-CSF therapy. The current standard of care in primary PAP (including aPAP) and some causes of secondary PAP [1]. Although inhaled GM-CSF therapyis an effective therapy for aPAP[2, 3], which, however, is off-label prescribing in PAP patients. That means the cost of inhaled GM-CSF therapy is completely at a patient's own expense instead of using Medicare reimbursement. Therefore, only a minority of patients accepted this therapy.

  1. Please indicate how you established the dose and time of therapy of GM-CSF. The literature has several proposals (PMID:32615994 and PMID:20167854)

Thanks for your comments. Because of the retrospective nature of this study, we could not intervene in the testing and treatment procedures. Therefore, the dose and time of inhaled GM-CSF therapy were prescribed by respiratory physicians, and we collected the information from the Hospital Information System (HIS). In general, the dose and time were adjusted based on one's symptoms, signs, and pulmonary manifestations on HRCT. As per the literature recommended[4, 5] and other reports[6-8], which showed the dose ranging from 125 μg/d (125 μg, q.d.) to 1000 μg/d (500 μg, b.i.d.) and the time ranging from 6 weeks to 65 months. We think the dose and time of inhaled GM-CSF therapy prescribed by respiratory physicians of West China Hospital were rational.

  1. Table 1. Please include data of FEV1, FVC, TLC, DLCO, PaO2, SpO2

Thank you for this valuable suggestion. We have added the data of pulmonary function testing (FEV1/ FVC) and artery blood gas analysis (PaO2 and SpO2) to Table 1. Since the three categories of data were not normally distributed, we performed Fisher's exact test on FEV1/FVC (≥ 70% vs. < 70%)[9], PaO2 (≥ 70 mmHg vs. < 70 mmHg) [10], and SpO2 (≥ 94% vs. < 94%)[11] between two groups, none of which showed a statistical difference. 

Because this is a retrospective real-world study, data missing is observed in the patients (because of the cost burden, patients' irregular return visits, medical database updating, etc.). This situation was also seen in other studies[10, 12, 13]. We did not include the data of TLC and DLCO because of serious data missing. Data are listed in the uploaded Excel file (named "patient information").

  1. Line 119. “Numeric data were evaluated for a normal distribution using…”; however, authors should use a normality test and after, apply parametric or non-parametric statistics.

Thanks for your comments. Numeric data in our study only include "age at diagnosis". We have performed normality tests on the data by using SPSS Statistics software (version 17.0), which turned out to follow the normal distribution.

  1. Table 2. “aPAP (N = 36) and sPAP (N = 20)”. How is it possible that you now have 20 subjects from the sPAP group if in Table 1 the authors indicate that they have N=14?

Thanks for your comments. We briefly explained the reason "Specimens of 18 aPAP patients and 10 sPAP patients offered qualified DNA for PCR amplification and sequencing, which had 36 and 20 HLA-DRB1 alleles, respectively" (Lines 167-169 in the original version; Lines 183-185 in the revised version). The sPAP group, in particular, had 10 out of 14 patients with qualified DNA for HLA-DRB1 genotyping. The 10 patients were genotyped for both HLA-DRB1 alleles, which constituted 10*2 = 20 HLA-DRB1 alleles. The same goes for the 18 aPAP patients that had 36 HLA-DRB1 alleles.

  1. Line 195: “Clinical remission (per either physicians' assessment or patient self-assessment)”. Please write the parameters used to evaluate clinical remission. 

Thanks for your comments. There was no specific parameter used to evaluate clinical remission per either physicians' assessment or patient self-assessment. We reviewed the Hospital Information System (HIS) that showed physicians' assessment was pulmonary manifestations on HRCT and the patient’s chief complaint. Besides, in December 2022, we performed a telephone survey to evaluate clinical remission per patient self-assessment by inquiring of patients about the symptoms (cough and/or sputum, dyspnea, etc.) after their visits.

  1. Line 195: How did you evaluate adherence to treatment?

Thanks for your comments. The patients' adherence was evaluated by their return visits. We have recorded the times of patients `s return visits in the uploaded Excel file (named "patient information").

  1. Line 196: Is the validity of clinical remission results the same if come from the physicians or the patient? Please add criteria and references to support your response.

Thanks for your comments. Yes, the assessment from the physician and the patient showed good consistency (yielding a concordance rate of about 80%). The details are listed in the uploaded Excel file (named "patient information").

  1. Line 242-244. Expand this topic in the discussion according to your results

 Line 242-244: "In our study, there was a tendency of HLA-DRB1*14:54 carriers to response better to inhaled GM-CSF than HLA-DRB1*08:03 carriers (57.1%) did. We hope carefully controlled prospective studies with a more diverse population and more diverse HLA alleles would allow a more conclusive assessment."

Thanks for your comments. We would like to expand the two sentences separately.

The allele HLA-DRB1*14:54, which was previously classified as HLA-DRB1*14:01, is a common allele in the Chinese population[14-16]. Studies have shown that HLA-DRB1*14:54 was significantly positively or negatively associated with some autoimmune diseases[16, 17]. Despite its prevalence in Chinese, HLA-DRB1*14:54 was only detected in the aPAP group in our study, supporting a positive relation with this pulmonary autoimmune disease (Lines 255-260 in the revised version).

The preliminary concept of the prospective study is to establish a phase 2a, multinational, multi-center, open-label study. Patients with aPAP were included after diagnosis (based on the combination analysis of medical history, HRCT manifestations, and serum levels of GM-CSF autoantibodies). Information including HLA-DRB1 genotyping, pulmonary function test, and blood gas analysis was collected on registration. As per the disease severity score (DSS)[10], patients were assigned to therapy groups with low, mediate, and high-dose inhaled GM-CSF (125-500 μg/ day). Rigorous and regular follow-up of patients on the self-reported symptoms, pulmonary function test, arterial blood gas analysis, and HRCT manifestation were performed during and after treatment.In addition, HLA-DRB1 information will also be collected from healthy people in the regions patients come from. The frequencies of different HLA-DRB1 alleles are to be compared between patients with aPAP and healthy population, and the remission rates in aPAP patients with different HLA-DRB1 alleles are to be analyzed.

  1. Discussion. The authors mentioned the association of HLA alleles in other populations; however, allele frequency depends on the ancestry population. Can you discuss this point?

We appreciate the valuable comment. We would like to discuss this point in two parts.

Firstly, after a particularly careful check of the clinical information, we can confirm that patients in the PAP group are not biologically related. They came from different provinces and cities in China and did not report PAP or similar diseases in their blood relatives based on their family history.

Secondly, Similar to Sakaue's findings in the Japanese population[18], our study confirmed that the frequency of HLA-DRB1*08:03 was 25% in aPAP, which constituted 50% of the tested aPAP patients. However, we did not observe statistical difference in the proportions of patients with HLA-DRB1*08:03 between aPAP and sPAP, indicating that HLA-DRB1*08:03 carriage was not exclusive to aPAP patients. The allele HLA-DRB1*14:54 is a common allele in the Chinese population[14, 15, 19]. Studies have shown that HLA-DRB1*14:54 was significantly positively or negatively associated with some autoimmune diseases. Despite its prevalence in Chinese [16, 17], HLA-DRB1*14:54 was only detected in the aPAP group in our study, supporting a positive relation with this pulmonary autoimmune disease (Lines 247-260 in the revised version).

References

  1. Trapnell, B.C., et al., Pulmonary alveolar proteinosis. Nat Rev Dis Primers, 2019. 5(1): p. 16.
  2. Iftikhar, H., G.B. Nair, and A. Kumar, Update on Diagnosis and Treatment of Adult Pulmonary Alveolar Proteinosis. Ther Clin Risk Manag, 2021. 17: p. 701-710.
  3. Tazawa, R., et al., Duration of benefit in patients with autoimmune pulmonary alveolar proteinosis after inhaled granulocyte-macrophage colony-stimulating factor therapy. Chest, 2014. 145(4): p. 729-737.
  4. Tian, X., et al., Inhaled granulocyte-macrophage colony stimulating factor for mild-to-moderate autoimmune pulmonary alveolar proteinosis - a six month phase II randomized study with 24 months of follow-up. Orphanet J Rare Dis, 2020. 15(1): p. 174.
  5. Tazawa, R., et al., Inhaled granulocyte/macrophage-colony stimulating factor as therapy for pulmonary alveolar proteinosis. Am J Respir Crit Care Med, 2010. 181(12): p. 1345-54.
  6. Trapnell, B.C., et al., Inhaled Molgramostim Therapy in Autoimmune Pulmonary Alveolar Proteinosis. N Engl J Med, 2020. 383(17): p. 1635-1644.
  7. Papiris, S.A., et al., Long-term inhaled granulocyte macrophage-colony-stimulating factor in autoimmune pulmonary alveolar proteinosis: effectiveness, safety, and lowest effective dose. Clin Drug Investig, 2014. 34(8): p. 553-64.
  8. Wylam, M.E., et al., Aerosol granulocyte-macrophage colony-stimulating factor for pulmonary alveolar proteinosis. Eur Respir J, 2006. 27(3): p. 585-93.
  9. Mirza, S., et al., COPD Guidelines: A Review of the 2018 GOLD Report. Mayo Clin Proc, 2018. 93(10): p. 1488-1502.
  10. Inoue, Y., et al., Characteristics of a large cohort of patients with autoimmune pulmonary alveolar proteinosis in Japan. Am J Respir Crit Care Med, 2008. 177(7): p. 752-62.
  11. Fanlo, P., et al., Efficacy and Safety of Anakinra Plus Standard of Care for Patients With Severe COVID-19: A Randomized Phase 2/3 Clinical Trial. JAMA Netw Open, 2023. 6(4): p. e237243.
  12. Salvator, H., et al., Pulmonary Alveolar Proteinosis After Allogeneic Hematopoietic Stem-Cell Transplantation in Adults: A French Société Francophone de Greffe de Moelle et Thérapie Cellulaire Survey. Chest, 2021. 160(5): p. 1783-1788.
  13. Goldstein, L.S., et al., Pulmonary alveolar proteinosis: clinical features and outcomes. Chest, 1998. 114(5): p. 1357-62.
  14. Yang, K.L., et al., New allele name of some HLA-DRB1*1401: HLA-DRB1*1454. Int J Immunogenet, 2009. 36(2): p. 119-20.
  15. He, Y., et al., HLA common and well-documented alleles in China. Hla, 2018. 92(4): p. 199-205.
  16. Luo, H., et al., The association of HLA-DRB1 alleles with antineutrophil cytoplasmic antibody-associated systemic vasculitis in Chinese patients. Hum Immunol, 2011. 72(5): p. 422-5.
  17. Saha, M., et al., Pemphigus vulgaris in White Europeans is linked with HLA Class II allele HLA DRB1*1454 but not DRB1*1401. J Invest Dermatol, 2010. 130(1): p. 311-4.
  18. Sakaue, S., et al., Genetic determinants of risk in autoimmune pulmonary alveolar proteinosis. Nat Commun, 2021. 12(1): p. 1032.
  19. He, Y., et al., HLA-DRB1*14:54:09 and -DRB1*14:54:10, were identified by next-generation sequencing in Chinese cord blood donors.Hla, 2021. 97(2): p. 166-169.

Reviewer 2 Report

The authors aims to retorspectively study 60 pathological patients diagnosed with PAP and devided into two groups regarding clinical diagnosis. The study is only descriptive providing information of common polymorphisms and one (HLA-DRB1*14:54) significantly different in aPAP. The results are clearly presented but the discussion part is lacking to provide meaningful explanation of the results in the light of reviewed literature. The conclusion part is not underlying the main findings and their importance. In the limitation of the study should be also commented the sample size and is it enough to make these conclusions. 

Author Response

Dear reviewer:

Thank you for your decision and constructive comments on my manuscript. We have carefully considered the suggestions and tried our best to improve and make some changes in the manuscript. We have made corresponding modifications in red font in the manuscript as you suggested.

Revision notes, point-to-point, are given as follows:

  1. The results are clearly presented but the discussion part is lacking to provide meaningful explanation of the results in the light of reviewed literature.

Thank you for your valuable comments. We added the following paragraph to the discussion part: "It is noteworthy that HLA allele frequency is related to the ancestry populations and geographical regions. HLA-DRB1*08:03 is an Asian-specific allele[1]. Similar to Sakaue's findings in the Japanese population, our study confirmed that the frequency of HLA-DRB1*08:03 was 25% in aPAP, which constituted 50% of the tested aPAP patients. However, we did not observe a statistical difference in the proportions of patients with HLA-DRB1*08:03 between aPAP and sPAP, which might be limited by the small sample size and low statistical power (1-β). At least,this indicated that HLA-DRB1*08:03 carriage was not exclusive to aPAP patients. The allele HLA-DRB1*14:54, which was previously classified as HLA-DRB1*14:01, is a common allele in the Chinese population. Studies have shown that HLA-DRB1*14:54 was significantly positively or negatively associated with some autoimmune diseases [2, 3]. Despite its prevalence in Chinese [3-5], HLA-DRB1*14:54 was only detected in the aPAP group in our study, supporting a positive relation with this pulmonary autoimmune disease. This is the first report revealing an underlying genetic predisposition of HLA-DRB1*14:54 carriers towards aPAP" (Lines 247-261 in the revised version).

  1. The conclusion part is not underlying the main findings and their importance.

Thank you for your valuable comments. We have modified the conclusion part as " Our study first reveals that population with HLA-DRB1*14:54 was subject to aPAP, and HLA-DRB1*14:54 might imply a response in aPAP patients to inhaled GM-CSF therapy. Our findings may contribute to the exploration of the underlying mechanism of aPAP. Further study is warranted to determine the association between HLA-DRB1*14:54 and aPAP" (Lines 297-301 in the revised version).

  1. In the limitation of the study should be also commented the sample size and is it enough to make these conclusions. 

Thank you for your valuable comments. We have performed statistical power (1-β) analysis by the G*Power software (version 3.1) on the differences in proportions of patients with HLA-DRB1*08:03, 14:54, and 15:01 between the two groups. The limitation part of our manuscript has been modified, and the following paragraph is added: "Second, the sample size for our study was limited. The statistical power (1-β) values of analyzing the differences in proportions of patients with HLA-DRB1*08:03, 14:54, and 15:01 betweentwo groups were 0.168, 0.763, and 0.610 respectively. Because of the non-significant P values (> 0.05) and low (1-β) values (< 0.75), the results from HLA-DRB1*08:03 and 15:01 analysis could not reflect true differences between the groups, and non-true negative results might occur. Nevertheless, the value for (1-β) in analyzing HLA-DRB1* 14:54 was 0.763 in the context of P < 0.05. We speculate that this value could reach 0.8 in an enlarged sample size" (Lines 283-290 in the revised version).

References

  1. Sakaue, S., et al., Genetic determinants of risk in autoimmune pulmonary alveolar proteinosis. Nat Commun, 2021. 12(1): p. 1032.
  2. Saha, M., et al., Pemphigus vulgaris in White Europeans is linked with HLA Class II allele HLA DRB1*1454 but not DRB1*1401. J Invest Dermatol, 2010. 130(1): p. 311-4.
  3. Luo, H., et al., The association of HLA-DRB1 alleles with antineutrophil cytoplasmic antibody-associated systemic vasculitis in Chinese patients. Hum Immunol, 2011. 72(5): p. 422-5.
  4. Yang, K.L., et al., New allele name of some HLA-DRB1*1401: HLA-DRB1*1454. Int J Immunogenet, 2009. 36(2): p. 119-20.
  5. He, Y., et al., HLA common and well-documented alleles in China. Hla, 2018. 92(4): p. 199-205.

Reviewer 3 Report

Thanks for the opportunity to review the manuscript by Mengqian Li and collaborators. In this brief report, the authors present a retrospective study of 60 patients pathologically diagnosed with PAP from 2019 to 2022, divided into the aPAP group or secondary PAP (sPAP) group according to their clinical information. PCR-SBT method was used for HLA-DRB1 genotyping in only 28 patients.

In this case-control study design, the main idea is interesting, but there are some concerns to be addressed before considering its suitability to be accepted for publication.

The authors initially described 69 patients, but only in 28, the HLA-DRB1 genotyping was done, reducing (still more) the limited sample size. Discussing is not enough; the authors should calculate the statistical power reached and include it in the discussion.

The authors should include the severity for patients in each group in Table 1.

The samples for histopathological procedures are comprehensible but limit the DNA genotyping. The study is relatively recent (2019 to May 2022). Why blood or saliva samples were not collected once the study was planned (to genotype HLA-DRB1)? 

Interestingly, regarding the GM-CSF therapy, 29 aPAP patients accepted inhaled therapy with doses ranging from 125 to 300 μg for from 2 weeks to 3 years, showing ample heterogeneity in its use. Please include the international guidelines applied in the treatment. Also, inhaled recombinant human GM-CSF therapy is the second-line treatment of aPAP; please describe if the patients are under the first line of treatment and how this modifies the results.

According to genotype results, HLA-DRB1*14:54 constituted the most frequent ones (19.4%, 7/36) and has never been reported in PAP. So, are patients in the PAP group biologically related (are they family?)? Please state it clearly.

It would be useful to compare the frequencies in the PAP patients (all) with healthy populational subjects in the same geographic area.

The first line in the conclusion is over the reach of the study due to the small sample size; it should be reformulated.

Author Response

Dear reviewer:

Thank you for your decision and constructive comments on my manuscript. We have carefully considered the suggestions and tried our best to improve and make some changes to the manuscript. We have made corresponding modifications in red font in the manuscript as you suggested.

Revision notes, point-to-point, are given as follows:

  1. The authors initially described 69 patients, but only in 28, the HLA-DRB1 genotyping was done, reducing (still more) the limited sample size. Discussing is not enough; the authors should calculate the statistical power reached and include it in the discussion.

Thank you for your comments. Because PAP is a rare diffuse lung disease, the sample size for our study was limited. In addition, PAP specimens available from the Department of Pathology, West China Hospital were mainly transbronchial biopsy specimens (56/60, 93%). It is logical because less than 10–20% of patients require surgical biopsy for confirmation[1]. Transbronchial biopsy specimens often provide a small amount of tissue that is subject to DNA degradation. Thus, cases with qualified DNA for HLA-DRB1 sequencing were limited.

We applied G*Power software, version 3.1 (www.gpower.hhu.de) to calculate the statistical power (1-β) of analyzing the differences in proportions of patients with HLA-DRB1*08:03, 14:54, and 15:01 between aPAP and sPAP group. Results are shown in the following table. Because of the non-significant P values and low (1-β) values (< 0.75) in HLA-DRB1*08:03 and 15:01 analysis, the results could not reflect true differences between the groups, and non-true negative results might occur. In particular, the value for (1-β) in analyzing HLA-DRB1* 14:54 was 0.763. We speculate that this value could reach 0.8 in an enlarged sample size (Lines 283-290 in the revised version). We have recognized this problem and plan to cooperate with respiratory physicians to expand the sample size and collect body fluid specimens in the following study

HLA-DRB1 alleles

Total (N = 28),

no. (%)

aPAP (N = 18),

no. (%)

sPAP (N = 10),

no. (%)

P value

Statistical power (1-β)

08:03

12 (42.9)

9 (50.0)

3 (30.0)

0.434

0.168

14:54

7 (25.0)

7 (38.9)

0 (0.0)

0.030

0.763

15:01

7 (1.8)

2 (11.1)

5 (50.0)

0.067

0.610

  1. The authors should include the severity for patients in each group in Table 1.

Thanks for this valuable suggestion. We have included the severity analysis in Table 1 as you suggested. Participants were assigned a PAP disease severity score (DSS) based on the presence of symptoms and the degree of reduction in PaO2 (both determined attheir first visit) determined with the individual breathing room air in the supine position as previously described[2]. DSS analysis (Mann-Whitney U test) did not show any difference between the two groups.

Besides, we measured the frequencies of "Critical or serious illness during hospitalization" in two groups based on their medical records, which have been added to Table 1. The results showed that this frequency was significantly higher in the sPAP group than in aPAP group (42.8% vs. 7.7%, P = 0.007). The sPAP group might have been associated with higher severity based on this data.  

  1. The samples for histopathological procedures are comprehensible but limit the DNA genotyping. The study is relatively recent (2019 to May 2022). Why blood or saliva samples were not collected once the study was planned (to genotype HLA-DRB1)? 

Thanks for your valuable suggestion. We retrospectively retrieved information and selected eligible PAP patients from the pathology database of the Department of Pathology, West China Hospital. By that time, patients had already been discharged from the hospital, and their blood samples were disposed of. Besides, not all patients came to the hospital regularly for outpatient follow-up visits. Therefore, it was difficult to ask patients for body fluid specimens. We have recognized this problem and plan to cooperate with respiratory physicians to expand the sample size and collect body fluid specimens in the following study.

  1. Interestingly, regarding the GM-CSF therapy, 29 aPAP patients accepted inhaled therapy with doses ranging from 125 to 300 μg for from 2 weeks to 3 years, showing ample heterogeneity in its use. Please include the international guidelines applied in the treatment. Also, inhaled recombinant human GM-CSF therapy is the second-line treatment of aPAP; please describe if the patients are under the first line of treatment and how this modifies the results.

Thanks for your comments.

For "Please include the international guidelines applied in the treatment": The current standard of care in primary PAP and some causes of secondary PAP (but not congenital PAP) is whole-lung lavage [3]. Although inhaled GM-CSF therapy is an effective therapy for aPAP, which have been supported by multiple studies[4, 5] and reviews[3, 6], inhaled GM-CSF therapy is still off-label prescribing in PAP patients, and there is no international guideline to follow at present. Patients were informed of this before treatment, and less than half of them were willing to afford this therapy that was at their own expense.

For "please describe if the patients are under the first line of treatment and how this modifies the results": In theory, patients who underwent the combined therapy (first-line + second-line therapy) should show better response. However, only 2 aPAP patients accepted the first-line of treatment (whole lung lavage) in our study, including one with HLA-DRB1*14:54/09:01 and the other one with HLA-DRB1*08:03/14:05. Since clinical remission with inhaled recombinant human GM-CSF therapy was observed in 100.0% (2/2) of HLA-DRB1*14:54/X carriers, 50.0% (2/4) of HLA-DRB1*08:03/X carriers, and 66.7% (2/3) HLA-DRB1*08:03/14:54 carriers, respectively (X did not include either 08:03 or 14:54), we speculate that the first-line therapy did not influence the results.

  1. According to genotype results, HLA-DRB1*14:54 constituted the most frequent ones (19.4%, 7/36) and has never been reported in PAP. So, are patients in the PAP group biologically related (are they family?)? Please state it clearly.

We appreciate the valuable comment. After a particularly careful check of the clinical information, we can confirm that patients in the PAP group are not biologically related. They came from different provinces and cities in China and did not report PAP or similar diseases in their blood relatives based on their family history. Besides, the allele HLA-DRB1*14:54 is common in the Chinesepopulation[7-9]. Studies have shown that HLA-DRB1*14:54 was significantly positively or negatively associated with some autoimmune diseases[10, 11]. Despite its prevalence in Chinese, HLA-DRB1*14:54 was only detected in the aPAP group in our study, supporting a positive relation with aPAP (Lines 247-261 in the revised version).

  1. It would be useful to compare the frequencies in the PAP patients (all) with healthy populational subjects in the same geographic area.

Thanks for your important advice. We agree that the comparison between PAP patients and non-PAP subjects can further reveal the association of HLA-DRB1 frequencies with PAP. We plan to compare the HLA-DRB1 frequencies in the PAP patients with healthy populational subjects, especially paying attention to the status of subjects in the same geographic area, in our next study.

  1. The first line in the conclusion is over the reach of the study due to the small sample size; it should be reformulated.

Thanks for your important advice. We have modified the first sentence in the conclusion into "Our study first reveals that population with HLA-DRB1*14:54 was subject to aPAP, and HLA-DRB1*14:54 might imply a response in aPAP patients to inhaled GM-CSF therapy "(Lines 297-299 in the revised version).

References

  1. Kumar, A., et al., Pulmonary alveolar proteinosis in adults: pathophysiology and clinical approach. Lancet Respir Med, 2018. 6(7): p. 554-565.
  2. Inoue, Y., et al., Characteristics of a large cohort of patients with autoimmune pulmonary alveolar proteinosis in Japan. Am J Respir Crit Care Med, 2008. 177(7): p. 752-62.
  3. Trapnell, B.C., et al., Pulmonary alveolar proteinosis. Nat Rev Dis Primers, 2019. 5(1): p. 16.
  4. Iftikhar, H., G.B. Nair, and A. Kumar, Update on Diagnosis and Treatment of Adult Pulmonary Alveolar Proteinosis. Ther Clin Risk Manag, 2021. 17: p. 701-710.
  5. Trapnell, B.C., et al., Inhaled Molgramostim Therapy in Autoimmune Pulmonary Alveolar Proteinosis. N Engl J Med, 2020. 383(17): p. 1635-1644.
  6. Jouneau, S., C. Ménard, and M. Lederlin, Pulmonary alveolar proteinosis. Respirology, 2020. 25(8): p. 816-826.
  7. He, Y., et al., HLA-DRB1*14:54:09 and -DRB1*14:54:10, were identified by next-generation sequencing in Chinese cord blood donors.Hla, 2021. 97(2): p. 166-169.
  8. Yang, K.L., et al., New allele name of some HLA-DRB1*1401: HLA-DRB1*1454. Int J Immunogenet, 2009. 36(2): p. 119-20.
  9. He, Y., et al., HLA common and well-documented alleles in China. Hla, 2018. 92(4): p. 199-205.
  10. Saha, M., et al., Pemphigus vulgaris in White Europeans is linked with HLA Class II allele HLA DRB1*1454 but not DRB1*1401. J Invest Dermatol, 2010. 130(1): p. 311-4.
  11. Luo, H., et al., The association of HLA-DRB1 alleles with antineutrophil cytoplasmic antibody-associated systemic vasculitis in Chinese patients. Hum Immunol, 2011. 72(5): p. 422-5.

Round 2

Reviewer 1 Report

Thank you for attending my observations

Reviewer 3 Report

Thanks for your conscientious review; I appreciate the time and commitment to the deep correction. You did a good work.